# Multifunctional Cotton Fabrics Obtained by Modification with Silanes Containing Esters of Phosphoric Acid as Substituents

**DOI:** 10.3390/ma14061542

**Published:** 2021-03-21

**Authors:** Marcin Przybylak, Michał Dutkiewicz, Karol Szubert, Hieronim Maciejewski, Szymon Rojewski

**Affiliations:** 1Poznan Science and Technology Park, Adam Mickiewicz University Foundation, Rubież 46, 61-612 Poznań, Poland; midu@amu.edu.pl (M.D.); maciejm@amu.edu.pl (H.M.); 2Centre for Advanced Technologies, Adam Mickiewicz University, Uniwersytetu Poznańskiego 10, 61-614 Poznań, Poland; 3Faculty of Chemistry, Adam Mickiewicz University, Uniwersytetu Poznańskiego 8, 61-614 Poznań, Poland; karolszu@amu.edu.pl; 4Institute of Natural Fibres & Medicinal Plants, Wojska Polskiego 71b, 60-630 Poznań, Poland; szymon.rojewski@iwnirz.pl

**Keywords:** cotton, flame retardants, organosilicon, coating, multifunctional, water repellent

## Abstract

The development of novel flame retardants for cotton textiles that form a stable layer on textile fiber is of high economical and practical relevance. A novel flame retardant fluorinated phosphoric acid esters modified silicone resins for cotton modification were synthesized. The investigated phosphoric acid esters based compounds were substituted by a fluorinated chain or ring, and alkoxysilyl groups. The presence of alkoxysilyl groups allowed the formation of bonds with cellulose, while derivatives of phosphoric esters reduced the flammability of fabrics. Additionally, the presence of fluoride in their structures affected the hydrophobic properties. Cotton fabrics were modified in a simple one-step process by dip-coating method. The flame retardant properties of modified textiles were examined by performing microcalorimetric analysis, thermogravimetry analysis, and measuring oxygen index. The hydrophobicity was evaluated by measuring the water contact angle. The modified fabrics were characterized by SEM-EDS (Scanning Electron Microscopy with Energy Dispersive Spectroscopy) analysis and surface morphology. As a result of the tests, multifunctional fabrics were obtained.

## 1. Introduction

Cotton fabrics are the most important branch of the natural textile industry. A strong ecological trend and growing awareness of consumers mean that the cotton fabric market is constantly growing. Cotton fabrics have many advantages such as soft touch, elasticity, and good mechanical properties. However, there is a problem in the flammability and hydrophilicity of cotton products. More and more stringent standards and safety requirements mean that it is necessary to look for new solutions to meet the expectations of producers and customers. To give fabrics new properties, such as being flame retardant or hydrophobic, they are subjected to modifications [1,2,3].

So far, natural textiles have usually been modified unidirectional to reach the flame retardancy or high hydrophobicity. A wide spectrum of flame retardant compounds has been used to protect fabric against fire. The most important group of compounds are halogen antipyrene; however, their use must be limited due to environmental protection [4,5,6,7,8,9]. There is, thankfully, a group of environmentally-safe fluorinated compounds. Such an example is the short-chain derivatives of fluorinated alcohols. Studies have shown that they do not pose a threat to humans and the environment and do not generate hazardous compounds during decomposition [10,11]. Phosphorus, as well as phosphorus and nitrogen-containing compounds which are effective in lowering flammability, has been also widely used [12,13,14,15,16,17,18,19]. Unfortunately, most of the investigated compounds are not able to bind permanently to the modified surface.

An interesting group of compounds used to reduce the flammability of natural fabrics is organosilicon compounds. The most popular form of organosilicon compounds is silica precursors, which are applied in the sol-gel process. These compounds can form multidimensional siloxane structures that insulate the fibers from the source of flames on the fabric surface [20,21,22,23,24]. Moreover, these siloxane networks can easily immobilize phosphorus or ammonium flame retardants [25,26,27,28].

Recent studies have focused on the synthesis of new organosilicon compounds having the ability to produce bonds with the cellulose and impart desired properties to surfaces subjected to the modification. There are literature reports on the synthesis of flame retardant compounds by substituting aminopropyltriethoxysilane with phosphorus and nitrogen derivatives. Derivatives such as 10-dihydro-9-oxa-10-phosphaphenanthrene-10-oxide and isophorone diisocyanate [29] or dimethyl hydroxymethylphosphonate and cyanuric chloride were used [30]. In another work, aminopropyltriethoxysilane has been substituted with phosphoric acid and urea [31]. Silicon and phosphorus-containing flame retardant for cotton textiles have been also synthesized from dimethyl phosphonate and methylvinyldichlorsilane [32]. Polysiloxanes have been also used for the synthesis of flame retardant compounds with reactive groups, eg., 4-bromobutoxy-terminated poly(dimethylsiloxane) substituted with piperazine and phosphorus oxychloride [33]. Another paper presents the synthesis of poly(4-iodobutoxy)methylsiloxane with N-methyl-3-(dimethoxydibenzyloxyphosphoryl)acrylamide to obtain a modifier for cotton fabric [34]. However, many of the developed methods have some disadvantages like a complicated impregnation process, low fire retardant effect, low washing resistance, or high add-on value necessary to obtain a satisfactory flame-retardant effect. The research group led by Hu S. et al. obtained a relatively low fire-retardant effect despite the use of high concentrations of the fire-retardant compound [29]. However, in the case of the research by Tian P et al., it was necessary to use an add-on modifier at the level of 30% to obtain a good fireproof effect of the fabrics [31].

In our previous studies, we reduced the flammability of fabrics both by using immobilization of ammonium phosphates in a silica sol and by synthesizing new compounds capable of binding with cellulose [35]. We have developed the synthesis of alkoxysilyl groups containing cyclophosphazenes used to modify cotton fabrics, reducing their flammability [36]. Moreover, we researched the synthesis of silanes, polysiloxanes, and silsesquioxanes containing alkoxysilyl and fluoroalkyl groups, the use of which for the fabric modification resulted in the preparation of superhydrophobic, and resistant for multiple washing processes cotton [37,38]. Experience gained in the synthesis of functionalized organosilicon compounds prompted us to the synthesis of new silanes containing three different functional groups. The research was aimed at obtaining compounds capable of producing covalent bonds with cellulose hydroxyl groups, granting hydrophobic and flame retardant properties to fabrics. As part of this study, three silanes with alkoxy groups, phosphoric acid esters, and fluorinated groups were synthesized. Cotton fabrics were modified with obtained compounds and then their fire- and waterproofing effect was examined.

## 2. Materials and Methods

The fabric used has a 145 g/m^2^ surface weight and was made in 100%, bleached before the modification process in a hydrogen peroxide bath cotton fabric. Tetraethoxysilane, triethoxysilane, octafluoropentanol were obtained from “Unisil” (Tarnów, Poland). Other reagents and solvents were purchased from Aldrich (Poznan, Poland) and used without any additional preparatory steps.

### 2.1. Synthesis of Organosilicon Flame Retardant Compounds

The synthesis of three types of organosilicon flame retardant compounds has been developed by our research group and has not been published previously.

#### 2.1.1. Synthesis of Organosilicon Derivatives of Fluorinated Phosphoric Esters (1 and 2)

The syntheses were carried out according to Scheme 1.

In a round-bottom flask equipped with a reflux condenser, a dropping funnel, a thermometer, and magnetic stirrer eugenol (60 mL; 0.38 mol), toluene (50 mL) and pyridine (30 mL; 0.38 mol) were placed, then the system was cooled to 0 °C. Next POCl_3_ (17.5 mL; 0.19 mol) was slowly added while maintaining the temperature below 10 °C. After the addition of the whole amount of POCl_3_, the system was kept at 0 °C for 2 h. After this time, the reaction mixture was allowed to stand overnight at room temperature. The resulting pyridine hydrochloride was then filtered off (precipitate was washed with toluene) and another portion of pyridine (20 mL, 0.24 mol) was added. The system was again cooled to 0 °C and octafluoropentyl alcohol (26.5 mL; 0.19 mol—to obtain compound **1**) or pentafluorophenol (20 mL; 0.19 mol) dissolved in 10 mL of toluene (to obtain compound **2**) were slowly added while maintaining the temperature below 10 °C. After the addition of the alcohol (or phenol), the temperature was kept at 0 °C for 2 h. After this time, the reaction mixture was allowed to stand overnight at room temperature, and then the mixture was subjected to filtration and excess of pyridine was evaporated under reduced pressure. In the next step, the obtained mid product (0.17 mol), toluene (50 mL), triethoxysilane (38 mL; 0.2 mol) and 2% Karstedt catalyst solution (21.5 µL; 1.88 × 10^−6^ mol of Pt) were placed together in a round-bottomed flask equipped with a thermometer, reflux condenser, and magnetic stirrer. The reaction was carried out at 90 °C for 2 h. Then, the excess of triethoxysilane was evaporated under reduced pressure. The colorless liquids were obtained with a yield of 85% and 83%, respectively.

The compound **1**:

^1^H NMR (C_6_D_6_, 298K, 300 MHz) ppm: 1.13–1.23 (Si-O-CH_2_-CH_3_, Si-CH_2_-CH_2_-); 3.08 (t, CH_2_); 3.43 (s, -O-CH_3_); 3.86 (q, Si-O-CH_2_-CH_3_); 5.32 (t, -P-O-CH_2_-CF_2_-); 5.63(t, -CF_2_H); 6.54–6.85 (CH phenyl).

^13^C NMR (C_6_D_6_, 298K, 75.5 MHz) ppm: 15.8 (Si-CH_2_-CH_2_-); 17.9 (Si-O-CH_2_-CH_3_); 23.2 (Si-CH_2_-CH_2_-); 40.1 (-CH_2_-); 55.6 (-O-CH_3_); 60.3 (Si-O-CH_2_-CH_3_); 113.5–150.8 (phenyl).

^29^Si NMR (C_6_D_6_, 298K, 59 MHz) ppm: −89.2

^31^P NMR (C_6_D_6_, 298K, 59 MHz) ppm: −1.3

The compound **2**:

^1^H NMR (C_6_D_6_, 298K, 300 MHz) ppm: 1.07–1.19 (Si-O-CH_2_-CH_3_, Si-CH_2_-CH_2_-); 3.12 (t, CH_2_); 3.39 (s, -O-CH_3_); 3.84 (q, Si-O-CH_2_-CH_3_); 6.54–7.33 (CH phenyl).

^13^C NMR (C_6_D_6_, 298K, 75.5 MHz) ppm: 15.8 (Si-CH_2_-CH_2_-); 17.9 (Si-O-CH_2_-CH_3_); 23.2 (Si-CH_2_-CH_2_-); 40.1 (-CH_2_-); 55.6 (-O-CH_3_); 60.3 (Si-O-CH_2_-CH_3_); 113.5–150.8 (phenyl).

^29^Si NMR (C_6_D_6_, 298K, 59 MHz) ppm: −90.1

^31^P NMR (C_6_D_6_, 298K, 59 MHz) ppm: −1.2

#### 2.1.2. Synthesis of Organosilicon Derivative of Phosphoric Diesters (3)

The synthesis was carried out according to Scheme 2.

The synthesis of compound 3 was carried out analogously to the synthesis described in Section 2.1.1. The 3-allyloxy-1,2-propanediol (20.8 mL; 0.19 mol) was used instead of eugenol, while water (1.7 mL; 0.19 mol) was added instead of octafluoropentanol (or pentafuorophenol). The product was obtained with a yield of 83%.

^1^H NMR (C_6_D_6_, 298K, 300 MHz) ppm: 0.93–1.14 (Si-O-CH_2_-CH_3_, Si-CH_2_-CH_2_-); 3.49–4.51 (Si-O-CH_2_-CH_3_, -O-CH_2_-, O-CH-).

^13^C NMR (C_6_D_6_, 298K, 75.5 MHz) ppm: 11.5(Si-CH_2_-CH_2_-); 16.8 (Si-O-CH_2_-CH_3_); 23.7 (Si-CH_2_-CH_2_-); 59.1 (Si-O-CH_2_-CH_3_); 69.3–74.1 (-O-CH_2_-, -O-CH-).

^29^Si NMR (C_6_D_6_, 298K, 59 MHz) ppm: −81.4

^31^P NMR (C_6_D_6_, 298K, 59 MHz) ppm: −1.9

### 2.2. Modification of Fabrics

Cotton fabrics were modified in two ways. In the one-step process, the bleached fabrics were impregnated with solutions of fire-retardant compounds 1, 2, 3 or in the two-step process the fabrics were initially mercerized.

#### 2.2.1. Cotton Fabric Impregnation with Organosilicon Flame Retardant Compounds (1, 2, and 3)

In a round-bottom flask 5 vol.% of isopropanolic solution of organosilicon flame retardant compound (1, 2, or 3) were placed; moreover, 5 vol.% of acetic acid and 5 vol.% of water were placed to the solution and hydrolysis were conduct for one hour at room temperature. The flask was equipped with a reflux condenser and a magnetic stirrer. The hydrolyzed solution was then transferred to laboratory trays where the fabrics were modified for half an hour. Then, the samples were squeezed and next dried for 60 min at 80 °C and cured for 3 min at 130 °C.

#### 2.2.2. Mercerization Process (M)

A 10% NaOH solution was prepared and then the bleached fabrics were soaked in it for 10 min. The fabrics were then rinsed intensively in water and dried.

#### 2.2.3. Washing Process

The modified fabrics were washed five times for 30 min at 40 °C according to the standard [39]. 

### 2.3. Analyses and Measurements

#### 2.3.1. NMR Spectroscopy

To make the ^13^C and ^1^H NMR spectra, a Bruker Ultrashield 300 MHz spectrometer (Bruker, Billerica, MA, USA) was used and the ^31^P and ^29^Si spectra were recorded on a Bruker Ascend 400 spectrometer (Bruker, Billerica, MA, USA). A CDCl_3_ solvent was used to make all spectra.

#### 2.3.2. Determination of the Amount of Modifiers Applied on Fabrics (Add-on)

The add-on value of cotton samples (A) was calculated according to the Equation (1). Where (*Wi*) was weighing a sample before washing and (*Wf*) was weighing a sample after washing. Analytical balance Ohaus PX224M/1 (OHAUS Europe GmbH, Nänikon, Switzerland) was used in the measurements.
(1)$$A=\frac{Wf-Wi}{Wi}*100$$

#### 2.3.3. Elemental Analysis of Coated Samples

The SEM-EDS (Scanning Electron Microscopy with Energy Dispersive Spectroscopy) technique was employed to determine the concentration of the C, O F, Si and P atoms on the surface of prepared cotton samples. Measurements were made using a Hitachi S-3500N (Hitachi Scientific Instruments, Schaumburg, IL, USA) (manufacturer, city, state, country) scanning electron microscope (SEM) equipped with a Thermo Scientific energy-dispersive X-ray detector (EDS) (Hitachi Scientific Instruments, Schaumburg, IL, USA).

#### 2.3.4. Microscale Combustion Calorimetry (MCC)

The flammability was assessed using a Pyrolysis-Combustion Flow Calorimeter (PCFC) produced by Fire Testing Technology Ltd. (East Grinstead, UK) West Susexx, (FTT). The heating rate (β) was 1 °C/s, the pyrolysis temperature range 75–750 °C, and the combustion temperature 900 °C. The flow was a mixture of O_2_/N_2_ 20/80 cm^3^/min and the sample weight 5–6 ± 0.01 mg. All measurements were repeated three times and the experimental error on HRR (heat release rate) was ±2% and the instrumental error on T was 1 °C and t was 1 s.

#### 2.3.5. Limiting Oxygen Index

The limiting oxygen index (LOI) were determined according to the standard [40]. Each fabric was tested three times and the mean value of the measurements was calculated. The experimental error was ± 0.5%.

#### 2.3.6. Thermogravimetric Analysis

Thermogravimetric analysis (TGA) of samples (9–10 mg) was evaluated using a TA Instruments Q50 TGA (TA Instruments, New Castle, DE, USA) thermobalance at a linear heating rate of 10 °C/min from room temperature to 700 °C under synthetic air (60 mL/min). The experimental error was 0.5% on weight and 1 °C on temperature.

#### 2.3.7. Studies of Surface Morphology

Surface morphologies of modified and unmodified samples were evaluated using Hitachi S-3400N scanning electron microscope (SEM) (Hitachi Scientific Instruments, Schaumburg, IL, USA). The samples were coated with a thin layer of gold before performing observations.

#### 2.3.8. Water Contact Angle (WCA) Measurements

The water contact angles (WCA) were measured using an automatic video contact-angle testing apparatus Krüss model DSA 100 Expert (Kruss, Hamburg, Germany). A 10 µL volume of water was applied onto the treated cotton fabrics. Each result is an average from the measurements of five drops. The experimental error was ±2%.

## 3. Results and Discussion

The synthesis of three new organosilicon flame retardant compositions was carried out in three steps. In the first and second steps, phosphororganic or fluorinated phosphororganic compounds have been synthesized, while in the third step the hydrosilylation with triethoxysilane was performed (Scheme 1 and Scheme 2).

All the obtained derivatives (1−3) contained alkoxysilyl groups susceptible to hydrolysis and condensation processes enabling their permanent binding to the substrate (cotton fibers). However, considering that all the derivatives (1−3) obtained have three alkoxysilyl groups, binding to the cotton is possible (for sterical reasons) only through one or two groups. The remaining groups condense together to form a siloxane layer on the surface of the modified fibers. In fact, partial condensation of the ethoxysilyl groups can be observed already during the isolation of products (1−3), which is confirmed by NMR analysis (especially ^29^Si). This partial condensation of compounds promotes the formation of fire-resistant coatings on the surface of modified fabrics. Thus, the observed pre-condensation effect is advantageous because it is possible to form a sealed fireproof coating on the surface of the modified fabrics. A lot of studies have been carried out to confirm the effect of condensation on the formation of a protective layer on the surface of the fibers and the improvement of flame-retardant properties [20,41,42]. In addition, our previous research has shown that as a result of condensation in the sol-gel process, a siloxane layer is formed which improves the fireproof effect [35,36].

The samples of bleached cotton fabrics were modified with both described compositions by pad dry cure method. At two-step processes, samples were subjected to the mercerization process before actual impregnation with obtained prepolymer solutions. The mercerization was carried out to facilitate the attachment of the modifier by activation of the surface of the textiles and to increase the number of hydroxyl groups capable to create covalent bonds with triethoxysilyl groups of organosilicon flame retardant compounds. All samples have been weighed before and after their modification. The add-on value was calculated on this basis (Table 1).

The data presented in Table 1 show that the value of add-on is at a similar level, regardless of the modifiers used. Fabrics modified with fluorinated phosphoric esters have a slightly higher add-on value due to the higher molecular weight. Furthermore, it can be seen that the cotton mercerization (applied in case of two-step impregnation process) resulted in an increased add-on value of all mercerized samples compared to those modified in a single-step process. It is worth noting that by applying a 5% solution of the modifier all of the add-ons are relatively low. Thanks to this, the modified fabrics retained their elasticity, soft-grip, and visual qualities. In some studies, to obtain a good fire retardant effect, it was necessary to use very high concentrations of modifiers, which had an impact on the high value of add-on (even 30%) [31].

Cotton samples impregnated with the obtained solutions in a single or two-stage process before and after washing were subjected to thermogravimetric analysis in the air atmosphere to assess the influence of applied modifications on the thermal stability of the fabric and durability of the impregnation. To facilitate the analysis and comparison of recorded TG curves presented in Figure 1, Figure 2, Figure 3 and Figure 4, the results of the TG and DTG analysis of impregnated samples including their temperatures of 1, 5, 10, and 20% mass loss (at which the tested samples lost 1, 5, 10, and 20% of their initial mass), temperatures of maximum decomposition rate (Tmax), measured decomposition rates at Tmax (Δm), as well as residue yields observed at Tmax and 700 °C, were presented in Table 2 and compared with the results obtained for unmodified cotton as a reference.

Based on the obtained TGA results (Table 2) and the course of the TG and DTG curves of cotton samples impregnated in a single-step process (Figure 1), it can be observed that the impregnation process resulted in a decrease in the thermal stability of the tested samples, regardless of the kind of the modification used, compared to pure cotton. In the case of samples 1 and 2, this was manifested by the appearance of an additional decomposition step in the temperature range 150–200 °C, most likely due to the partial breakdown of the C-F bonds present in the structure of applied impregnates and the release of HF. For all tested samples (1, 2, and 3), the T_onset_ temperature of the beginning of the main decomposition step measured as the intersection of the mass baseline and the tangent to the TG curve at the point of maximum rate of mass loss decreased from 335 °C for neat cotton to 325, 316, and 280 °C for samples 1, 2, and 3, respectively. At the same time, the change in the slope of the TG curves of the modified samples and the significant reduction in the intensity of the signals on their DTG curves presented in Figure 1b (especially for sample 3) indicate a reduction in the rate of main decomposition step related to the formation of volatiles, combustible and non-combustible species yielding aliphatic char in dehydration and cellulose depolymerization processes [43,44,45]. While for samples 1 and 2 no significant changes were observed in the char yield formed after the first decomposition step (at 380 °C) compared to the reference sample, its yield produced during the degradation of sample 3 was significantly higher and amounted to 37% compared to 14% for the pure cotton sample. The impregnation of the samples with the obtained preparations also influenced the second stage of decomposition of the tested samples related to the conversion of aliphatic char into aromatic structures, with the slow evolution of water, methane, carbon mono and dioxide [36,37,38] by its spread-in-time compared to the reference sample. This is evidenced by the extension and flattening of the TG and DTG curves in the temperature range from 380 to 600 °C. This effect is visible for sample 2 and especially sample 3.

Subsequently, to evaluate the effect of the mercerization process on the impregnation efficiency samples impregnated in two-step process were also subjected the thermogravimetric analysis and its results were compared with those measured for cotton samples impregnated in single-step impregnation. The results of the TG and DTG analysis of 1M, 2M, and 3M samples showed a further reduction in the temperatures of the beginning of the first stage of decomposition by about 5 to 10 °C, a further reduction in the rate of the first decomposition stage and an increase in the char yield formed after the first decomposition step, and a significant spread in time of the second decomposition step, compared to the not mercerized samples. It was especially pronounced for samples 1M and 2M (char yield higher by about 10%).

To assess the effect of the impregnation method on the durability of protective coatings obtained cotton samples impregnated in single and two-step process and subjected to the five-time washing were also subjected the thermogravimetric analysis and its results were compared with those measured for cotton samples impregnated in single or two-step impregnation before washing. The results presented in Figure 3 showed that the repeated washing process clearly weakens the observed effects. This is the result of the partial leaching of the applied coatings as a result of hydrolysis of the C-O-Si bonds formed during impregnation between cellulose and silyl derivatives of phosphate esters, which is also evidenced by the change in weight of the washed samples compared to freshly impregnated ones. Nevertheless, the course of the TG and DTG curves of the **1**W, **2**W, and **3**W samples differ from those of pure cotton. An additional decomposition step in the temperature range of 150–200 °C is still clearly observed, as well as a reduction in the decomposition rate of the discussed samples in the first and second stage, illustrated by a decrease in the intensity of the DTG curves peak in Figure 3b and the flattening of the TG curves in the temperature range 380–600 °C (Figure 3a).

As mentioned above, the use of the two-stage impregnation process improved its efficiency. This effect is also confirmed by the results of TG and DTG analyzes of 1MW, 2MW, and 3MW samples subjected to two-stage impregnation and multiple washing processes, shown in Figure 4. Although the changes in the course of the curves compared to these of reference sample are not as clear as in the case of their counterparts before washing, as a result of the previously mentioned partial degradation of C-O-Si bonds, they are still significantly more pronounced than in the case of samples impregnated without the mercerization process after washing. The reduction of the maxima on the DTG curves in the range of 300–400 °C and their shift towards lower temperatures (Figure 4b) as well as the spread overtime of the second decomposition stage of in the temperature range 380–600 °C (Figure 4a,b) are much clearer. For better understanding of the applied impregnation method and modifier structure effect on the thermal stability of measured samples and the durability of obtained coatings, additional plots showing the superimposed TG and DTG curves of cotton samples impregnated with all modifiers (1, 2, and 3) in a single and two-stage process before and after washing were presented in the supplementary materials file for the manuscript (Appendix A).

Subsequently, all prepared samples of bleached cotton fabric (raw and modified in single and two-step impregnation) before and after the washing process were subjected to pyrolysis-combustion flow calorimetry (PCFC) measurements to assess the flame retardancy of the fabricated coatings. The obtained results of PCFC analysis are summarized in Table 3 where 1, 2, 3—type of modifier (Scheme 1 and Scheme 2), M—mercerization process, W—washing process.

Results of the PCFC analysis obtained for cotton fabric samples modified with all preparations in a single and two-step processes influenced the reduction of the cotton textile flammability compared to the raw fabric as reference. The flame retardant effect of modified fabrics depended strongly on the structure of the compound used as well as the impregnation process applied.

As can be seen in Figure 5a and Table 3. temperatures of maximum heat release rate (T_PHRR_) of all measured samples (1, 2 and 3) were shifted toward lower temperatures form 392 °C for raw cotton textile to about 317 °C for samples 1 and 2 and 365 °C for sample 3 with a simultaneous decrease in the value of maximum heat release rate (P_HRR_) from 270 W·g^−1^ to 106, 112 and 203 W·g^−1^ for samples 1, 2, and 3, respectively. In the case of samples 1 and 2, the single-step impregnation resulted also in the significant (over 45%) reduction in total heat release (THR) value form 12 kJ·g^−1^ observed for reference sample to about 6.5 kJ·g^−1^. THR value measured for sample 3 was almost unchanged compared to the raw cotton. This may be explained by the presence of two aromatic rings the structure of derivatives 1 and 2, as well as fluoroalkyl or fluorophenyl groups. It is well known that the presence of aromatic rings in compounds increases their thermal stability and promotes the formation of carbonaceous char during their decomposition, acting as an insulating surface layer. Thanks to this mechanism, heat conduction inside the sample is slowed down and the evolution of flammable gases is reduced. Charring is the most common mode of action in the condensed-phase [5,46,47]. From the other side, the presence of fluorinated groups in the structure of impregnates 1 and 2 also can influence the fire-retardant mechanism of the modifying compositions [5,46]. Halogen compounds disrupt the free radical mechanism during combustion, which is responsible for providing heat to continue this process. The halides react in the gas phase to form hydrohalides upon combustion, which form more stable and less reactive free radicals, thus reducing the amount of heat needed to maintain the flame and simultaneously reducing the amount of oxygen in the burning zone [48,49]. Another scenario of phosphorus-halide synergy involves the action of halides as a blowing agent to foam the yielded char rather than to operate in the gas phase as a free radical scavenger [50]. Regardless of the mechanism of phosphorus-halides synergy, the understanding of which was not the aim of this work and certainly requires more complex research the presence of fluorine atoms in the structure of derivatives 1 and 2 certainly effectively influenced their flame-retardant effect.

Results of the PCFC analysis and the assumption of the occurrence of phosphorus-halide synergy seem to be consistent with the results of TG analysis.

Similarly, the results of TG analysis and also the PCFC tests confirmed the positive effect of the mercerization process on the efficiency of cotton samples impregnation as a result of the cotton surface activation and increase in the number of active centers (hydroxyl groups) and subsequently the reduction of their flammability. As can be observed in Figure 5b. Above-mentioned trends (shift of the T_PHRR_ toward lower values and decrease in P_HRR_ and THR values) are more pronounced for samples 1M, 2M, and 3M subjected to the two-step impregnation. T_PHRR_ temperatures were shifted to the 309, 304, and 359 °C, P_HRR_ values were decreased to 70, 73 and 142 Wg^−1^ for 1M, 2M, and 3M samples, respectively.

The shift of the T_PHRR_ toward lower values and decrease in P_HRR_ and THR values are favorable effects. This phenomenon is caused by the formation of char instead of flammable gases at lower temperatures, which results in the release of less energy. As a result of this process, less flammable gases and more residues were created, which lowered the flammability of fabrics.

To evaluate the durability of the coatings formed the samples subjected to the five-time washing process were also tested by the PCFC technique. Results of the PCFC measurements of washed samples 1W, 2W, and 3W as well as their modified in two-step process analogs (1MW, 2MW, and, 3MW samples) are presented in Figure 6 as temperature-resolved heat release rate curves and in Table 3.

Although the partial deterioration of the flammability parameters may be observed after five-time washings due to the hydrolytic degradation of protective coatings the flame retardant properties were not completely lost and T_PERR_, P_HRR_, and THR indices for all samples were still better than those observed for the raw cotton sample. It can be also observed that mercerized fabrics modified in two-step process show lower flammability after five washes than their non-mercerized analogous. The better flame retardancy parameters observed for samples modified with compounds 1 and 2 subjected to the five washing cycles may be explained by the presence of fluorinated groups in the structure of compounds used for their impregnation. These substituents of a hydrophobic nature can protect the surface of modified textile against water and partially prevent leaching resulting from the hydrolysis of C-O-Si bonds between the modifier and cellulose. A similar effect was observed in the case of TG analysis. The lack of complete resistance to washing of the applied modifications is a disadvantage of the described process. Flame retardant fabric compositions should be wash resistant to effectively protect people and infrastructure from fire. However, scientists all over the world are struggling with this problem and it has not been successfully solved so far.

The limited oxygen index (LOI) was also determined for all modified fabrics. The measurement results are presented in Figure 7.

The results presented in Figure 7 show that all applied modifications resulted in an increase in the oxygen index. LOI results are correlated with those obtained on PCFC and the best fire protective effect was achieved by modification with prepolymers 1 and 2. All samples obtained in the two-stage process (mercerized) have an oxygen index high enough to be considered as difficult to ignite. The value of the limited oxygen index for 1M and 2M samples was as high as 30.8%.

To additionally confirm the modifications, the SEM-EDS analysis was performed. Concentrations of C, O, F, Si, F, and P atoms (expressed in wt.%) in all modified samples is presented in Table 4.

The data presented in Table 4 confirm the presence of the modifier in all samples. The presence of silicon and phosphorus in all samples indicates that the fabrics have been successfully modified with the obtained preparations. Analysis of fabrics modified with prepolymers containing fluorinated phosphoric ester groups also showed the presence of fluorine. In the case of sample 1, the fluorine content is higher than in sample 2, which is a consequence of the higher content of this element in precursor 1. Samples 3 and 3M contain twice as much phosphorus as other fabrics, which is caused by the presence of two phosphorus atoms in the structure of the precursor used. In addition, there is a tendency that mercerized samples contain more elements indicating the presence of a used compounds. This indicates that mercerization has caused the attachment of a larger amount of modifier to the surface of the fabric. Analysis of the samples after washing shows that the modifiers remained on the surface of the fibers. However, the content of elements decreased, which indicates a partial leaching of compounds. The greatest reduction of element content can be observed for samples 3 and 3M. This may be due to the lack of fluorinated substituents in compound 3. As already mentioned, the presence of hydrophobic fluorinated groups protects the compounds against water access and leaching.

In order to study the morphology of modified fabrics, SEM analysis of their surfaces was performed (Figure 8).

By analyzing the SEM images shown in Figure 8, presence of the modifier layer can be observed on the surface of all impregnated samples regardless of the type of modifier used (1, 2, or 3) and the type of impregnation process applied. The micrographs show a layer of evenly distributed polymer surrounding each fiber. In the case of 2M and 3M samples impregnated in a two-step process, a few agglomerates are visible on their micrographs. The mercerization applied in the case of two-step impregnation procedure resulted in the deposition of a thicker layer of deposited polymers (samples 2M and 3M) compared to their non-mercerized analogous (samples 2 and 3). Washed fabrics modified with prepolymer 2 retained a modifier layer on the fibers surface (samples 2W and 2MW). SEM images confirmed earlier observations that the presence of fluorinated groups protect the deposited coatings from leaching.

To further confirm the effect of fluoroalkyl or fluorophenyl groups on the surface properties of modified cotton samples, the hydrophobicity of the studied organosilicon flame retardant compounds was determined by measuring the water contact angle of unwashed samples and those subjected to the five-time washing process. The values of water contact angles of the silane-modified samples are presented in Figure 9.

The modification of natural fabrics by organosilicon flame retardant prepolymers (1, 2, and 3) resulted in hydrophobization of the fabrics as reflected by values of water contact angles, which in all cases were above 120°. After washing, the hydrophobicity remained relatively stable in the case of samples modified by fluorinated phosphoric ester groups containing prepolymers. Fabrics modified with a composition 1 containing fluoroalkyl chains are characterized by the highest hydrophobicity (WCA of 135°), which is caused by the highest content of fluorine and its favorable distribution on the chain. The sample modified with a composition 2 containing fluorophenyl substituents is slightly less hydrophobic. Fabrics protected with a prepolymer 3 also exhibit hydrophobic properties, which is a consequence of the hydrocarbon and siloxane chains present in their structure. However, the hydrophobic effect disappears as a result of washing, which confirms the largest leaching of the coating devoided of fluorinated functional groups. Fluorinated groups are responsible for the hydrophobic effect of the modified fabric. These substituents of a hydrophobic nature can protect the surface of modified textile against water and partially prevent leaching as a result of C-O-Si bonds between the modifier and cellulose hydrolysis. The lack of fluorinated groups resulted in washing the modifier from the surface of the fabrics in the washing process. Therefore, the water-repellent effect after washing disappeared. In Figure 9 it can be seen that the fabric mercerization process resulted in an increase in the water contact angle values and hydrophobicity of modified samples. This is a consequence of the activation of the fiber surface and an increase in the number of active hydroxyl groups ready to bind with alkoxysilyl groups of flame retardant prepolymers. As a result, these compounds obtained better spatial organization of fluorinated groups, which translated into higher hydrophobicity of the samples modified with their use.

## 4. Conclusions

The proposed synthetic procedure, based on the nucleophilic substitution and subsequent hydrosilylation processes, enabled the obtaining of prepolymers containing phosphoric acid silyl ester groups and their fluorinated analogs in their structure as precursors of multifunctional protective coatings of natural fabrics. The results of thermal, flammability, and surface tests of cotton fabric samples subjected to a single or two-stage impregnation with the obtained prepolymers confirmed their effectiveness as precursors of hydrophobic and flame-retardant coatings. It was also shown that the cotton fabric mercerization process (used in the case of two-stage impregnation) is of key importance for its effectiveness and significantly contributed to the intensification of the observed effects (increased hydrophobicity and reduced flammability of the tested samples). Particularly good results as the reduction in the total amount of heat released during the burning process by over 50%, the reduction in the maximum heat release rate by about 75%, shortening the time to ignition by almost 90 min compared to raw cotton, as well as increase in the limiting oxygen index to over 30 and the increase in the water contact angle to about 130° were obtained for the samples modified with prepolymers 1 and 2 containing fluoroalkyl or fluorophenyl groups in their structure. Moreover, the results of discussed tests of samples subjected to the multiple washing processes showed that despite the partial degradation of the coatings produced as a result of hydrolysis of the Si-O-C bonds between the obtained prepolymers and cellulose, formed during the modification of cotton, they retained their hydrophobic properties and reduced flammability. At the same time, the test results showed that cotton samples modified with prepolymer 3, deprived of fluorinated functional groups, were characterized by significantly higher flammability, lower water contact angle, and hydrolytic stability, as evidenced by further deterioration of the discussed parameters measured for samples subjected to five washing cycles.

The obtained test results seem to confirm the synergy effect between the halides and phosphorus present in the structure of the precursors used for the production of protective coatings. Observed phosphorus-halide synergy can involve the action of halides in the gas phase as a free radical scavenger or the condensed phase as a blowing agent increasing the insulating char volume. As was mentioned, regardless of the mechanism of phosphorus-halides synergy, the understanding of which was not the aim of this work, and certainly requires more complex research, the presence of fluorine atoms in the structure of derivatives 1 and 2 unambiguously influenced observed flame-retardant effects.

## Data Availability

Data available on request due to privacy restrictions. The data presented in this study are available on request from the corresponding author. The data are not publicly available due to the lack of availability of a suitable repository.

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
