# Peer review of "Multifunctional Cotton Fabrics Obtained by Modification with Silanes Containing Esters of Phosphoric Acid as Substituents"

_materials, 2021, doi:10.3390/ma14061542_

Round 1

Reviewer 1 Report

This paper presented the synthesis of three silane coupling agents, which were modified with fluorinated phosphoric acid esters. Their performance as flame retardants for cotton textiles were characterized comprehensively. Moreover, the effects of the mercerization process on the flame retardant performance of modified cotton fabrics were also studied.  There are both scientific and technological interests in this work.    The comments are summarized as follows:

  1. It is stated that there is partial condensation of silane compounds in Page 6 and is expected to promote the formation of fire resistant coatings on cotton fabrics. More discussion and proof on this need to be given in detail.
  2. Scale bar for SEM image of cotton in Fig 8 needs to be provided.   
  3. Error bars are suggested to input in Fig 7.
  4. English needs to be polished, as well as some mistyped words, such as ‘wasched/unwasched’ in Figure 9.

Author Response

Dear Reviever

I wish to extend my sincere thanks for the insightful analysis of our manuscript and very pertinent remarks and comments. We have taken  comments into consideration and corrected the manuscript following your suggestions. I hope the revised version meets the requirements for publication in Matrials.

  1. It is stated that there is partial condensation of silane compounds in Page 6 and is expected to promote the formation of fire resistant coatings on cotton fabrics. More discussion and proof on this need to be given in detail.

Thank you so much for this valuable suggestion. We have supplemented the Results and Discussion with suggested explanations.

  1. Scale bar for SEM image of cotton in Fig 8 needs to be provided.   

Thanks for this comment. All photos were taken to the same scale. The scale bar is located at the bottom of the lower photos.

  1. Error bars are suggested to input in Fig 7.

Thank you for this remark. We have added error bars to the figure 7.

  1. English needs to be polished, as well as some mistyped words, such as ‘wasched/unwasched’ in Figure 9.

Thank you for this remark. We corrected the manuscript and figures linguistic errors.

Reviewer 2 Report

In this work, the novel flame retardants fluorinated phosphoric acid esters modified silicone resins for cotton were reported. The proposed chemicals show good flame retardant properties. I recommend this manuscript for publication after minor revision.

  1. The author should be checked the English spelling in figure 9    

                unwasched, wasched

     2. In the case of chemical 3, the water contact angle drastically decreased after washing. Why did the water contact angle decreases. 

Author Response

Dear Reviever

I wish to extend my sincere thanks for the insightful analysis of our manuscript and very pertinent remarks and comments. We have taken  comments into consideration and corrected the manuscript following your suggestions. I hope the revised version meets the requirements for publication in Matrials.

  1. The author should be checked the English spelling in figure 9    

                unwasched, wasched

Thank you so much for this valuable suggestion. We improved the legend in Figure 9.

  1. In the case of chemical 3, the water contact angle drastically decreased after washing. Why did the water contact angle decreases. 

Thank you very much for this question. Compounds 1 and 2 have fluorine-containing groups in their structure. These groups are responsible for the hydrophobic effect of the modified fabric. These substituent of a hydrophobic nature can protect the surface of modified textile against water and partially prevent leaching as a result of C-O-Si bonds between the modifier and cellulose hydrolysis.  The lack of fluorinated groups resulted in washing the modifier from the surface of the fabrics in the washing process. Therefore, the water-repellent effect after washing has disappeared.

Reviewer 3 Report

Manuscript ID: materials-1126751

Title: Multifunctional cotton fabrics obtained by modification with silanes containing esters of phosphoric acid as substituents

The presented work is on sol-gel finishing with fluoro- and phosphor-modified alkoxy silanes to impart flame-retardance and hydrophobicity to cellulosic fabrics. The work is interesting and will be of interest to the journal readers, but it is requested the following issues be addressed.

Lines 42-44: “However, there is a group of environmentally safe fluorinated compounds. Such an example is the short-chain derivatives of fluorinated alcohols. Studies have shown that they do not pose a threat to humans and the environment and do not generate hazardous compounds during decomposition.”

Please cite reference(s) in support of this assertion.

Lines 68-70: “However, many of the developed methods have some disadvantages like a complicated impregnation process, low fire retardant effect, low washing resistance, or high add-on value necessary to obtain a satisfactory flame-retardant effect.”

Please give details on what is complicated about the reported impregnation processes, what is a “low” fire retardant effect, and what is a “high” add-on value.

Please cite standard deviations (or a similar measure) to illustrate the variance in results shown Tables 1-4 and Figures 7-9.

It is difficult to interpret the results shown in Table 2. For example, what do the values 61.4, 175.1, 320.5, 344.9 represent for Sample 1. Please clarify.

Lines 285-286: “it can be observed that the impregnation process resulted in a decrease in the thermal stability of the tested samples”.

From the TGA plots, it appears that the untreated cotton degrades at lower temperatures than the treated samples, but the statement quoted above suggests otherwise. Please clarify.

Lines 305-306: “This is evidenced by the expropriation of the 305 TG and DTG curves in the temperature range from 380 to 600 °C.”

What does expropriation mean?

Please include at least one plot with the TGA profiles of control, treated, mercerized+treated, mercerized+treated+washed, so that the reader may obtain a better understanding of the differences.

In Table 3, please include a legend explaining the meaning of the abbreviations. It will be easier for readers to understand the values.

Paragraph beginning from line 489: There are frequent references to a modifier, and it is unclear what is being referred to.

There is a typo in the spelling of “washing” in figures.

It is difficult to follow and understand the paragraphs discussing TGA and MCC data. Too many concepts appear to be mixed together, without a clear distinction of when one point of discussion ends and the other begins. Please rephrase.

Author Response

Dear Reviever

I wish to extend my sincere thanks for the insightful analysis of our manuscript and very pertinent remarks and comments. We have taken  comments into consideration and corrected the manuscript following your suggestions. I hope the revised version meets the requirements for publication in Matrials.

Lines 42-44: “However, there is a group of environmentally safe fluorinated compounds. Such an example is the short-chain derivatives of fluorinated alcohols. Studies have shown that they do not pose a threat to humans and the environment and do not generate hazardous compounds during decomposition.”

Please cite reference(s) in support of this assertion.

Thank you so much for this valuable suggestion. We have supplemented the introduction with suggested publications.

Moreover, fabrics modified with compounds described in publication (Przybylak, M.; Maciejewski, H.; Dutkiewicz, A.; DÄ…bek, I.; Nowicki, M. Fabrication of superhydrophobic cotton fabrics by a simple chemical modification. Cellulose 2016, 23, 2185-2197.) have been certified as "Safe for Humans". The conducted studies have shown that the compounds do not show toxicity and are completely safe for humans.

Lines 68-70: “However, many of the developed methods have some disadvantages like a complicated impregnation process, low fire retardant effect, low washing resistance, or high add-on value necessary to obtain a satisfactory flame-retardant effect.”

Please give details on what is complicated about the reported impregnation processes, what is a “low” fire retardant effect, and what is a “high” add-on value.

Thank you for this remark. We have added literature references to the said sentence in the introduction. Fabric modification with plasma is often very effective, but requires specialized equipment and is usually a multi-stage process. In publication 29, despite the use of high concentrations of the modifier, the reduction of HRR was low. However, in publication 31 it was necessary to use an add-on of 30% to obtain a good fireproof effect.

Please cite standard deviations (or a similar measure) to illustrate the variance in results shown Tables 1-4 and Figures 7-9.

 The accuracy of the measurements made is given in section 2.3. Analyses and measurements for the relevant analytical methods. However, for better readability, this information has been also added as footnotes to tables or error bars where appropriate.

It is difficult to interpret the results shown in Table 2. For example, what do the values 61.4, 175.1, 320.5, 344.9 represent for Sample 1. Please clarify.

 According to the table heading, mentioned values are the temperatures at which the tested sample loses 1, 5, 10 and 20% of its initial mass, respectively. These values are commonly given to facilitate the analysis and comparison of TG curves.

Lines 285-286: “it can be observed that the impregnation process resulted in a decrease in the thermal stability of the tested samples”.

From the TGA plots, it appears that the untreated cotton degrades at lower temperatures than the treated samples, but the statement quoted above suggests otherwise. Please clarify.

 Unfortunately, we cannot agree with this opinion. Both the results of the TG and DTG analysis presented in Table 2 as well as the course of the TG and DTG curves shown in Figures 1-4 suggest a decrease in thermal stability of modified cotton samples. This is evidenced by the decrease in the 5, 10 and 20% mass loss temperatures and the temperature of the maximum decomposition rate observed for all modified cotton samples compared to the values measured for the pure cotton one. In the TG curves this phenomenon is visible in the form of a shift of the inflection of the curves towards lower temperatures, while in the DTG curves in the form of a shift of the signal maximum towards lower temperatures.

Lines 305-306: “This is evidenced by the expropriation of the 305 TG and DTG curves in the temperature range from 380 to 600 °C.”

What does expropriation mean?

 The word "expropriation" has been used incorrectly as a result of the autocorrect function instead of "extesion". The sentence was corrected in the manuscript to: „This is evidenced by the extension and flattening of the TG and DTG curves in the temperature range from 380 to 600 °C.”

Please include at least one plot with the TGA profiles of control, treated, mercerized+treated, mercerized+treated+washed, so that the reader may obtain a better understanding of the differences.

 Thank you for your valuable suggestion. In order not to unnecessarily increase the volume of the manuscript and not to duplicate the presented data, the superimposed TG and DTG curves of cotton samples impregnated with all modifiers (1, 2, and 3) in a single and two-stage process before and after washing were presented in the supplementary materials file for manuscript. A sentence directing the reader to the additional content has been added in the manuscript at the end of the discussion of the thermogravimetric analysis results. Added sentence: „For better understanding of the applied impregnation method and modifier structure effect on the thermal stability of measured samples and the durability of obtained coatings, additional plots showing the superimposed TG and DTG curves of cotton samples impregnated with all modifiers (1, 2, and 3) in a single and two-stage process before and after washing were presented in the supplementary materials file for the manuscript (Fig. S1-S3).”

In Table 3, please include a legend explaining the meaning of the abbreviations. It will be easier for readers to understand the values.

Thank you very much for this suggestion. We have added the explanation of symbols at table 3. A legend explaining the meaning of the abbreviations used in the table 3 has been added as footnotes for table: ttI - time to ignition, TPHRR - temperature of maximum heat release rate, PHRR - maximum heat release rate, THR - total heat release, ΔTHR – change of total heat release compared to reference sample (cotton)

Paragraph beginning from line 489: There are frequent references to a modifier, and it is unclear what is being referred to.

 The mentioned fragment of the text has been re-edited for better clarity.

By analyzing the SEM images shown in Figure 8, the presence of modifier layer can be observed on the surface of all impregnated samples regardles of the type of modifier uded (1, 2, or 3) and the type of impregnation proces applied. The micrographs show a layer of evenly distributed polymer surrounding each fiber. In the case of 2M and 3M samples impregnated in two-step proces a few agglomerates are visible on their micrographs. The mercerization applied in the case of two-step impregnation procedure resulted in the deposition of a thicker layer of deposited polymers (samples 2M and 3M) compared to their non-mercerized analogous (samples 2 and 3). Washed fabrics modified with prepolymer 2 retained a modifier layer on the fibers surface (samples 2W and 2MW). SEM images confirmed earlier observations that the presence of fluorinated groups protect the deposited coatings from leaching.

There is a typo in the spelling of “washing” in figures.

 Thank you so much for this valuable suggestion. We improved the legend in Figure 9.

It is difficult to follow and understand the paragraphs discussing TGA and MCC data. Too many concepts appear to be mixed together, without a clear distinction of when one point of discussion ends and the other begins. Please rephrase.

The sections discussing the results of the TG and MCC analysis have been improved to facilitate their better readability. All the amendments introduced were marked in yellow in the revised manuscript.

Reviewer 4 Report

The manuscript reports on the use of three novel organosilicon compounds as flame retardant and water repellant agents for cotton modification. The manuscript is generally well written, and introduction focuses very well the topic of the manuscript. However, some issues have to be addressed before its publication. What it is really missing in the manuscript is the lack of a critical review of the presented results with respect to the state of the art. Second, the scarce washing resistance shown by compound 3 is not well explained and cannot be justify by only considering the absence of fluorine in the chemical structure.    

Other minor issues are listed below:

Materials and methods section: in the thermogravimetric analysis paragraph, authors should complete the description by adding definitions and formula for Tonset, Tmax, and decomposition rate.

Lines 241-243: please check this sentence: the third step is always related to the hydrosilylation in both scheme 1 and 2, or not?

Line 247: carrier or substrate?

Table 2: the first part (left side) of this table is not clear, considering also that it is not discussed in the text. The authors should highlight its meaning.

Line 287: impregnation??

Line 476: what do the authors intend with the term “modifier”?

Lines 493-495: it is impossible to conclude the formation of a thicker layer from the SEM images reported in the text.

Author Response

Dear Reviever

I wish to extend my sincere thanks for the insightful analysis of our manuscript and very pertinent remarks and comments. We have taken  comments into consideration and corrected the manuscript following your suggestions. I hope the revised version meets the requirements for publication in Matrials.

As suggested, we have added a critical paragraph in line 441. Moreover, we have written additional explanations for the disappearance of the hydrophobic effect for compound 3 in line 520.

Other minor issues are listed below:

Materials and methods section: in the thermogravimetric analysis paragraph, authors should complete the description by adding definitions and formula for Tonset, Tmax, and decomposition rate.

Thank you for your attention, the above-mentioned definitions are listed under Table 2.

Lines 241-243: please check this sentence: the third step is always related to the hydrosilylation in both scheme 1 and 2, or not?

Thank you very much for this valuable comment, we have corrected this mistake in the manuscript.

Line 247: carrier or substrate?

Thank you very much for this question, we have corrected this sentence in the manuscript.

Table 2: the first part (left side) of this table is not clear, considering also that it is not discussed in the text. The authors should highlight its meaning.

Thank you very much for this valuable comment. According to the table heading, mentioned values are the temperatures at which the tested sample loses 1, 5, 10 and 20% of its initial mass, respectively. These values are commonly given to facilitate the analysis and comparison of TG curves. Moreover we have edited the text below the table.  

Line 287: impregnation??

Thank you very much for this comment, we have re-edited this sentence.

Line 476: what do the authors intend with the term “modifier”?

Thank you very much for this comment, we have re-edited this sentence.

Lines 493-495: it is impossible to conclude the formation of a thicker layer from the SEM images reported in the text.

Thank you very much for this comment, we have re-edited this sentence.

Round 2

Reviewer 1 Report

It could be accepted.

Author Response

Thank you very much for your help, comments and approval of our manuscript.

Reviewer 3 Report

Title: Multifunctional cotton fabrics obtained by modification with silanes containing esters of phosphoric acid as substituents

Revised version.

With reference to my comment on lines 68-70, the authors have only added references. I request that the authors add text to the manuscript to clearly explain what the authors consider complicated and what is a high add on value. And how the author developed method of impregnation under reflux conditions for 30 min is advantageous over previously reported methods – especially if industrial-scale treatments are considered.

Please add this information in the manuscript text “According to the table heading, mentioned values are the temperatures at which the tested sample loses 1, 5, 10 and 20% of its initial mass, respectively. These values are commonly given to facilitate the analysis and comparison of TG curves.”

“Unfortunately, we cannot agree with this opinion. Both the results of the TG and DTG analysis presented in Table 2 as well as the course of the TG and DTG curves shown in Figures 1-4 suggest a decrease in thermal stability of modified cotton samples.”

Agreed. It was my mistake.

“In order not to unnecessarily increase the volume of the manuscript and not to duplicate the presented data, the superimposed TG and DTG curves of cotton samples impregnated with all modifiers (1, 2, and 3) in a single and two-stage process before and after washing were presented in the supplementary materials file for manuscript.”

Unfortunately, I could not open the provided file. Please ensure it is readable.

Line 292: The authors have a typo where “king” is written instead of “kind”.

Author Response

With reference to my comment on lines 68-70, the authors have only added references. I request that the authors add text to the manuscript to clearly explain what the authors consider complicated and what is a high add on value. And how the author developed method of impregnation under reflux conditions for 30 min is advantageous over previously reported methods – especially if industrial-scale treatments are considered.

Thank you very much for this comment, we have supplemented the introduction with the information mentioned.

When it comes to the method of modifying the fabrics, there has been a mistake. Fabrics are modified in litter boxes without reflux. We have edited this sentence to make it clearer.

Please add this information in the manuscript text “According to the table heading, mentioned values are the temperatures at which the tested sample loses 1, 5, 10 and 20% of its initial mass, respectively. These values are commonly given to facilitate the analysis and comparison of TG curves.”

 Thank you very much for this comment, we have added this information to the manuscript above table 2. 

“In order not to unnecessarily increase the volume of the manuscript and not to duplicate the presented data, the superimposed TG and DTG curves of cotton samples impregnated with all modifiers (1, 2, and 3) in a single and two-stage process before and after washing were presented in the supplementary materials file for manuscript.”

Unfortunately, I could not open the provided file. Please ensure it is readable.

 We apologize for this error, we have attached the correct file.

Line 292: The authors have a typo where “king” is written instead of “kind”.

Thank you for your attention, we have corrected this mistake.

Reviewer 4 Report

The authors have revised the manuscript and answered to my questions.

In Line 292 there is a typo: king 

Author Response

(The authors gave the same response as above.)
